# The Impact of Parental Education on Schoolchildren’s Oral Health—A Multicenter Cross-Sectional Study in Romania

**DOI:** 10.3390/ijerph191711102

**Published:** 2022-09-05

**Authors:** Ramona Dumitrescu, Ruxandra Sava-Rosianu, Daniela Jumanca, Octavia Balean, Lia-Raluca Damian, Aurora Doris Fratila, Laurentiu Maricutoiu, Adrian Ioan Hajdu, Roxanne Focht, Mihaela Adina Dumitrache, Constantin Daguci, Mariana Postolache, Corina Vernic, Atena Galuscan

**Affiliations:** 1Translational and Experimental Clinical Research Centre in Oral Health, Department of Preventive, Community Dentistry and Oral Health, University of Medicine and Pharmacy “Victor Babes”, 300040 Timisoara, Romania; 2Faculty of Dental Medicine, Ludwig-Maximilian-University Munich, Goethestraße 70, 80336 München, Germany; 3Department of Psychology, West University of Timisoara, 300223 Timisoara, Romania; 4Oral Health and Community Dentistry Department, Faculty of Dental Medicine, Carol Davila University of Medicine and Pharmacy, 020021 Bucharest, Romania; 5Department of Oral Health, Faculty of Dentistry, University of Medicine and Pharmacy, 200585 Craiova, Romania; 6Department of Program Implementation and Coordination, Romanian Ministry of Health, 010024 Bucharest, Romania; 7Discipline of Computer Science and Medical Biostatistics, “Victor Babes” University of Medicine and Pharmacy, 300041 Timisoara, Romania

**Keywords:** oral health, caries, children, parents’ education

## Abstract

The present study is part of the first national oral health survey for children in Romania. The aim of this study was to determine caries prevalence in correlation with the level of the parents’ education, preventive behavior, and socioeconomic parameters in 11–14-year-old schoolchildren in Romania. A cross-sectional epidemiological survey was designed and conducted in 2019–2020. The sampled children were selected from 49 schools distributed in rural and urban areas of Romania, including its capital. Data were collected using the Oral Health Questionnaire for Children developed by the World Health Organization and described in the WHO Oral Health Surveys—Basic Methods, 5th edition, 2013, after positive informed consent. To express prevalence and severity of carious lesions, International Caries Detection and Assessment System (ICDAS) criteria were recorded in school for 814 schoolchildren (388 boys and 426 girls) aged between 11 and 14 years old (mean age 12.29 ± 0.6). Elements regarding the specificity of the child (gender, age, and parental education) were tabulated against preventive behavior. The parents’ education was correlated with three clinical indices in order to assess the existence or lack of certain significant differences among schoolchildren in Romania. In terms of correlation between the mother’s education and preventive behavior, results showed a significant positive correlation in case of dental check-ups (r_s_ = 0.08 *, *p* < 0.05), brushing (r_s_ = 0.02 **, *p* < 0.01), and use of different types of dental hygiene aids (r_s_ = 0.06 **, *p* < 0.01) and a negative correlation with tooth pain or discomfort (r_s_ = −0.01 **, *p* < 0.01). A statistically significant positive relationship was highlighted between the mother’s education and the presence of restorations (r_s_ = −0.09 **, *p* < 0.01). Regarding the father’s education, there was a positive relationship with oral hygiene behavior (r_s_ = 0.18 **, *p* < 0.01) but a negative relationship with the D_3_T index (r_s_ = −0.18 **, *p* < 0.01). In conclusion, there was a strong correlation between the parents’ education, preventive behavior, and oral health status of Romanian schoolchildren.

## 1. Introduction

Dental caries, although almost entirely preventable, are one of the most common afflictions worldwide, affecting people of all age groups and socioeconomic strata, having serious burdens for those affected. With 60–90% of schoolchildren affected by tooth decay worldwide, it represents a serious public health problem proven to affect the functional, psychological, and social dimensions of a child’s well-being [1]. Since oral diseases have been included in the non-communicable diseases (NCD) spectrum, the fact that oral health is an integral part of general health, sharing major common risk factors with other NCSs, was also highlighted [2]. Besides causing infection and pain, dental caries can significantly impact children’s quality of life, being associated with impaired development, reduced school attendance, and behavioral disorders. Experiencing pain, low self-esteem, difficulties chewing, or dealing with reduced school attendance from a young age due to oral health problems can significantly affect children’s well-being [3,4]. Previous studies have described a direct link between the prevalence of early childhood caries (ECC) and poor oral health-related quality of life [5]. Thus, oral afflictions during childhood can have a negative impact on the life of children, as well as their parents. Invasive dental treatments can be very expensive, creating financial burdens on the families, and as well, the emotional distress caused by pain, both for the child and for the parent, should be considered. By applying preventive strategies such as frequent visits at the dental office for professional preventive care and good maintenance of oral hygiene, these disadvantages can be avoided.

Although dental caries is preventable in early stages, according to the World Health Organization (WHO), their prevalence continues to rise in most low and middle-income countries unable to provide proper oral health services. Since treatment for oral health conditions is expensive and usually not included in universal health coverage, there is a profoundly uneven distribution of oral health services both between and within countries. These circumstances are most prevalent within marginalized communities and disadvantaged populations. For most of the twentieth century, dental caries was considered an affliction of the economically developed countries, with low prevalence in developing countries. Most developed countries and many non-industrialized countries are now well below the World Health Organization’s (WHO) goal of less than three decayed, missing, or filled teeth per 12-year-old child.

Preventive behavior plays a key role in ensuring proper oral health for children. The dimension of prevention behavior concerns aspects such as oral hygiene, access to dental services, proper care of teeth and gums, use of appropriate cleaning objects, and use of toothpaste. These elements related to the specificity of oral care, as well as visits to the dentist, must be carried out periodically from an early age so as to prevent the occurrence of health problems in the body. Poor oral hygiene leads to tooth decay, which if not treated properly can lead to tooth loss. Children’s adoption of consistent behavioral habits begins at home with their parents, especially mothers; they have a great influence on the child’s oral health behavior [6]. For younger children especially, the role of parents and primary caregivers is critical for caries control [7]. Within this context, maternal/parental education, attitudes and beliefs, and other psychosocial factors represent important mediators and moderators of parents’ oral health behaviors on behalf of their children [8,9,10,11,12]. Studies have reported that parental education has a direct impact on children’s oral health [13,14]. Low-education families do not pay enough attention to dental care measures and regular preventive visits to a dental professional, and this results in the development of dental caries [15].

To the best of our knowledge, there are insufficient national centralized studies reporting the status of children’s oral health in Romania. Data regarding oral health status can raise community awareness, as well as help reorient dental treatments from invasive therapy to prevention [16]. Currently, Romania does not benefit from a drinking water fluoridation system, and unfortunately, there are no official data regarding the concentration of fluoride in drinking water. The toothpastes usually commercialized in Romania contain fluoride according to the recommendations in force.

The dental care system in Romania is entirely private. Some of the dental practices are also in contractual relations with the National Health Insurance House of Romania, on the basis of which the mentioned institution settles a fixed monthly amount of money for dental treatments. This amount covers dental services for both adults and children on a first-come, first-served basis, there being no difference between the number of services offered to adults and children. Even if preventive dental procedures are offered and practiced, based on this contract, procedures such as professional brushing, fluoridation, and sealing fissures and pits, as well as instruction on brushing and oral care methods, are limited to a very small number of children who have access and who benefit from free preventive dental services based on the fact that there is a monthly limit.

The present study represents a part of the first national oral health survey for children aged 6–7 and 11–14 years, using the WHO guidelines for basic surveys.

Several determinants of health, such as social and economic circumstances, exist apart from the health sector altogether. Therefore, studies collecting and reporting epidemiological data can be used to not only increase awareness among the community but also provide evidence for policy-making strategies and targeted development.

The aim of the present study was to assess caries prevalence and to identify the correlation between the parents’ level of education, preventive behavior, and socioeconomic parameters with the oral health status in 11–14-year-old children in Romania.

## 2. Materials and Methods

### 2.1. Study Design and Sample Selection

A cross-sectional epidemiological survey, “Romanian Oral Health Survey”, was designed and conducted in 2019–2020 with the endorsement of the World Health Organization (WHO) and the support of the Ministry of Health and in conformity with the General Data Protection Regulations 2018. The study methodology was approved by the relevant local (school authority), regional (school inspectorate), national authorities (Ministry of Health from Romania—approval no. 3411/05.04.2018 and Ministry of Education—approval no. 1573/12.03.2019), and the University of Medicine and Pharmacy “Victor Babes”, Timisoara, Romania (no. 29/28.09.2018). Two age groups of schoolchildren were included in the survey, with an average age of 6 years, and 12 years of age being the inclusion factor. The pattern of dental eruption is the one that motivated the choice of this age, namely the fact that at 6 years, the first permanent molar erupts, and at the age of 12, the period of eruption of permanent teeth ends. The 6-year-old age group was studied separately due to the peculiarities of the mixed dentition (both temporary and permanent teeth being present), with the present study focusing on the 12-year-old age group, which is the period of young permanent dentition. In Romania, the schooling system starts from grade 0 (around the age of 6 for children: primary school (grades 0 to 4), followed by middle/secondary school/gymnasium (grades 5 to 8), and then either high school (grades 9 to 12) or vocational/trade school (grades 9–13). Higher education (university) can be achieved only after children graduate high school. Regarding secondary education, the class includes on average 22 schoolchildren but no less than 10 and no more than 26. The selection of schools and classes that were examined was made in accordance with the WHO procedure for Oral Surveys in order to ensure a representative sample. The sampled children were selected from 49 schools, from 4696 schools that fit our inclusion criteria, distributed in rural and urban areas from the counties of Romania, including its capital. At every school, the involvement in the study was made available to the whole class. Although the participation rate was not recorded in detail, it was possible to estimate an inclusion of at least 90% of all children in these 49 schools. To ensure the randomization and stratification of the sample, for each county, according to the number of pupils registered for the 8th grade National Exam in every Romanian school, the total number of children was determined and expressed as a percentage share. This process involved the evaluation of publicly available information from 4696 schools. Consequently, the percentage share for each county was used to estimate the number of children who would need to be included. Thus, the obtained number was divided according to the locality type (i.e., urban or rural). In this way, the target number of necessary evaluations was determined. The examinations were carried out until reaching the required number of children according to the sample size calculation, regardless of the number of children in a class.

For selecting an urban and a rural school for each county, the randomization function of MS Excel was used. A total of 49 schools were selected, all distributed in rural as well as urban areas. School level predictors also included the County Developmental Index [17], which is a sociological index that combines county-level variables: life expectancy at birth, educational level, average age of the adult population, average living area, number of privately owned cars to 1000 inhabitants, and average gas consumption per household. The aggregation of all these different variables was done using factor analysis, and the scores were relevant for assessing the county’s workforce potential and the county’s economic potential.

The Oral Health Questionnaire for Children developed by the World Health Organization and described in the WHO Oral Health Surveys—Basic Methods, 5th edition, 2013 [18] was used for self-assessment. Two independent translators translated the English version of the questionnaire, and the differences were settled in a face-to-face meeting. Furthermore, two experts were consulted, one in educational psychology and another in developmental psychology, to ensure the readability of the original Romanian version of the Questionnaire for Children. It includes questions regarding general information, age, frequency of and reason for dental check-ups, oral hygiene behavior, use of fluoridated toothpaste, and dietary behavior, as well as the educational level of the parents/caregivers. According to Romanian education law no. 1/2011, compulsory general education includes primary education (until 4th grade), secondary education (until 8th grade), and the first 2 years of upper secondary education (vocational), which is entirely supported by the state [19]. Additionally, the questionnaire includes inquiries about pain related to teeth, frequency of dental visits, tooth cleaning habits, and consumption of sugary foods/drinks. Questionnaires were sent to the schools by postal services, submitted to the children by the teachers one week prior to the clinical examination, and filled in by the children in collaboration with their parents/caretakers at home. The parents/caretakers had to fill informed consent forms, which were gathered by the examiners on the day of examination (Figure 1).

### 2.2. Clinical Examination

Survey teams majoring in preventive dentistry in dental schools joined in the 2019 survey. Examination was conducted by ten trained and calibrated investigators. The necessary training and calibration of the participating dentists was performed before the data collection. The calibration procedure was carried out by examining 21 subjects who were not included in the final sample. Kappa statistic was used to test the investigator reliability, displaying an inter-examiner kappa ranging from 0.74–0.86 and intra-examiner kappa ranging from 0.81–0.92. The kappa coefficient for fillings was consistently in the excellent range, while for lesion severity, it was in the good to excellent range. No re-examination of the children included in the present sample was conducted because the clinical investigation was done continuously during one school year, and the examiners had good results in the calibration stage. During data collection, children were examined in a classroom or any available room, with the teacher present during the procedure, using special front LED lighting sources commonly used in dentistry and examination kits. Considering the environment where the examinations were performed, radiographs were not used for diagnosis. Inclusion criteria were children without chronic diseases who were not on medication, irrespective of sex, race, and socioeconomic status, and who were cooperative. The exclusion criterion was children wearing braces for fixed orthodontic treatment.

Each participant came to the examination with the informed consent and questionnaire signed by the parents/caregivers. ICDAS criteria were used to classify visual caries lesion severity, and the ICCMS Guide for Practitioners and Educators [20] was used to classify the presence of filling material on all surfaces of permanent and primary teeth. Dental plaque or food debris was removed using cotton rolls. For every surface analyzed, ICDAS codes were documented on a specific chart. ICDAS codes 1 and 2 were recorded as “A”, as air drying was not possible. The examination was carried out from one tooth to the adjacent tooth. A tooth was considered present if any part of it was visible in the oral cavity. If permanent and primary tooth occupied the same tooth space, the status of the permanent tooth was recorded. The collected data were coded, introduced in examination charts, later saved electronically, and were annexed to the questionnaire and consent form for each participant.

### 2.3. Structure of the Questionnaire

The questionnaire consists of 15 items that make up the two dimensions of types of behavior (prevention and food), radically influencing oral health and dental care. Preventive behavior was assessed by presence of pain or discomfort within the last 12 months, frequency of dental check-ups, reason for dental check-ups, frequency of oral hygiene, and aids for oral hygiene.

### 2.4. Data Analysis

In a concrete way, for a better understanding of the factors that determine the prevention behaviors of children in relation to oral health, the scientific approach aimed at correlating elements regarding the specificity of the child (gender, age, and parental education). The parents’ education level was also correlated with three clinical indices in order to determine the existence or lack of certain significant differences among schoolchildren in Romania: MT = missing teeth, the number of teeth with ICDAS codes 97/98 indicates the teeth that have been extracted due to caries; RT = restoration (number of surfaces that have a restoration/sealing): as the teeth can have 4–5 surfaces each, this index can take high values; and D_3_T = number of surfaces with a caries code greater than 3 (cavitated enamel). The codes range from 0 to 6–7, and the codes from 3 upwards indicate the presence of enamel lesions (the rest are incipient lesions, without enamel breakdown).

The statistical processing of the data obtained through the questionnaires was done by Statistical Package for the Social Sciences (SPSS) 23 version for Windows. Spearman’s rank correlation, a nonparametric test (r_s_ or rho), was used for quantitative and ordinal variables. It indicates how strongly two variables are monotonously related. The Pearson correlation was used for the relation between county development index and the D_3_T, MT, and RT indices. It was also applied to measure the strength of a linear association between two variables, the clinical status and the county development index, in order to see how this social index influences the children’s oral health status. Statistically significant values were set at *p* < 0.05.

## 3. Results

A total of 814 schoolchildren (388 boys and 426 girls) aged between 11 and 14 years old, with an average age of 12.29 ± 0.6, participated in the study as shown in Table 1.

Among the personal parameters of the participants in the study, one of the factors considered was the education of the mother, and of the father, as a determining factor of the children’s oral health. Regarding the mother’s education from the studied sample, 122 persons declared that they graduated gymnasium (8th grade), 234 graduated high school (12th grade), and 181 graduated university. In terms of the father’s education, 911 persons declared that they graduated gymnasium, 269 of them graduated high school, and 149 persons graduated university (Table 2).

### 3.1. Parents’ Education Correlated to Preventive Behavior

According to the results, there was a significant negative relationship between the mother’s education and the frequency of toothache and discomfort reported by children in the last 12 months (r_s_ = −0.11, *p* < 0.01). Regarding the connection between the mother’s education and the frequency of visits to the dentist reported by the children, there was a positive and significant relationship (r_s_ = 0.08 *, *p* < 0.05), which means the higher the level of education of the mother was, the higher the number of children’s visits to the dentist were. The relation between the mother’s education and the frequency of tooth brushing in children was positive and significant (r_s_ = 0.02 **, *p* < 0.01), so the higher the mother’s education, the higher the frequency of tooth brushing in children. A significant and positive relationship could also be seen between the use of several oral hygiene aids (mouthwash, electric toothbrush, and mouth shower) and the level of education of the mother (r_s_ = 0.06 **, *p* < 0.01), so children whose mothers had a better education used various tools when cleaning their teeth.

Regarding the relation between the father’s education and the preventive behavior, this was partially supported by the results of the Spearman rho correlation on this sample. A significant and positive association could be seen between the father’s education and the frequency of tooth brushing in children (r_s_ = 0.18 **, *p* < 0.01), so the higher the level of education of the father, the higher the frequency of cleaning children’s teeth (Table 3).

### 3.2. Parents’ Education Correlated to Clinical Status

ICDAS codes higher than 3 (caries into dentin) were computed as dentinal caries “D_3_T” (number of carious tooth surfaces), missing teeth due to complex carious lesions as “MT”, and the number of restored tooth surfaces as “RT”. The value of the DMFT index determined for the sample was 2.93 ± 2.70.

Regarding the relation between the mother’s education and the DMFT index, based on the Spearman rho correlation analysis, there was a significant positive link between the mother’s education level and the RT index (r_s_ = 0.09 **, *p* < 0.01), where the higher the education of the mother, the higher the RT index in children. Regarding the relation between the D_3_T index and the mother’s education, the data reported a significant negative correlation (r_s_ = −0.19 **, *p* < 0.01) such that the higher the mother’s education, the lower the D_3_T in children. The mean DMFT was correlated significantly and negatively with the level of education of the mother (r_s_ = −0.09 **, *p* < 0.01), which means the higher the level of education of the mother, the lower the DMFT index in children.

Regarding the components of the DMFT index taken separately and the father’s education, a significant and negative link between the D_3_T index and education (r_s_ = −0.18 **, *p* < 0.01) could be observed; thus, fathers who had a higher level of education had children with a lower D_3_T index. There was a significant negative relationship between DMFT and the level of education of the father (r_s_ = −0.10 **, *p* < 0.001), so the higher the level of education of the father, the lower the DMFT index (Table 4).

### 3.3. Preventive Behavior Correlated to Clinical Status

In terms of the correlation between preventive behavior and clinical status, the results of the nonparametric correlation analysis, Spearman rho, show in Table 5 that there was a positive significant link between the DMFT index and the frequency of dental pain/discomfort reported in the last 12 months (r_s_ = 0.11 **, *p* < 0.01). The RT index correlated significantly and positively with the frequency of tooth brushing r_s_ = 0.07 *, *p* < 0.05, so we can say that those who brushed their teeth more frequently had a higher RT index. The MT index was positively and significantly associated with the frequency of dental pain in the last 12 months, r_s_ = 0.09 **, *p* < 0.01 (children who reported a lot of pain in the last 12 months had a higher MT index). There was also a significant negative link between the frequency of dental check-ups and the MT index, r_s_ = −0.07 *, *p* < 0.05, so children who have gone to the dentist many times in recent years had a low MT index. The D_3_T index was correlated with most of the registered habits, making it the strongest index.

This was significantly positively correlated with the frequency of dental pain complaints in the last 12 months, r_s_ = 0.15 **, *p* < 0.01, meaning that those with many caries had a high frequency of pain discomfort in the last 12 months. A significant negative relationship existed between D_3_T and the frequency of tooth brushing, r_s_ = −0.01 **, *p* < 0.01, meaning that those who had a high number of cavitated lesions also reported a low frequency of tooth brushing (Table 5).

### 3.4. Clinical Status Correlated to County Development Index

The MT index was correlated with the county development level (the 2011 LHDI development index). The results showed that there was no significant relationship between the incidence of prevention behavior and the level of development of the county where the child came from. There was a significant relationship between the D_3_ index and the level of development of the county from which the respondent came, with the level of significance being 0.044 and the Pearson correlation coefficient being −0.070 *. Additionally, the tests showed that there was no statistically significant relationship between the RT index and the development index of the county where the child came from (Table 6).

## 4. Discussion

In Romania, few oral epidemiological studies have been carried out during recent years, and systematic data on oral health behavior of children are scarce. The present survey provides information as regards the influence of the parents’ education on the preventive comportment of their children and the oral health status in 11 to 14-year-old urban and rural schoolchildren. The survey was conducted on a national scale, and therefore, the data are representative for the country in pure statistical terms.

Given that oral health issues are extremely important among schoolchildren, determining the patterns that influence the prevalence of caries is essential for their prevention efforts.

According to the HBSC cross-national study in 2001–2002, across 27 European countries, Israel, Canada, and USA, the geographical differences in DMFT were substantial, with levels ranging from less than 1.0 in Denmark (0.9), Netherlands (0.8), Sweden (0.9), Switzerland (0.9), England (0.9), and Wales (0.9) to over 3.5 in Lithuania (3.6), Poland (3.8), and Ukraine (4.4). The DMFT of most Eastern and Central European countries was above 1.6 [21]. The literature data show that European countries have a lower DMFT index, with 0.5 in Germany (2014) [22], 0.6 in the Netherlands (2012), 0.7 in the UK (2011), 0.7 in Spain (2014), and 0.8 in Sweden (2011) [23,24] compared to 3.13 in Romania (2020) [25].

Based on the results from a previous Romanian study [26], four categories of predictors explain the variance in the dentinal caries index and fillings/restoration index: personal characteristics of the respondents, hygienic behavior, consumption behavior, and living area characteristics. In a small local Romanian study that took place in Cluj-Napoca in 2017 on a sample of 650 schoolchildren with a mean age of 15.3 ± 2.8 years, research showed that there was high percentage of Romanian adolescents (40.6%, N = 264) who missed the regular dental yearly check-up and visited the dentist when they are in pain, both in females and males, without significant differences [25]. As recently reported in Romania, looking at every category which sums up the DMFT index, the relationship between the incidence of missing teeth (the MT component) and preventative oral health-related behavior is influenced by the individual characteristics of the child (gender, age, and parents’ education) [27].

Parental attitudes towards children’s oral health depend on their education. The study carried out by Rajab et al. reported that better-educated parents cared more about their children’s oral health [28]. Reduced parental education is regarded to be one of the main factors leading to poor oral health in children. Moreover, parental educational levels were reported to be directly associated with family socioeconomic status [13]. The results reported in Italy [29] showed that the children of parents with a high level of education had fewer caries than the subjects of parents with a low level of education. Children coming from families with highly educated parents had a higher chance of getting restorative treatment [27]. Data from the literature showed that the low level of education of parents was related to the high risk of caries among schoolchildren [30]. Low maternal schooling showed a borderline association with higher levels of dental caries [31]. The higher the parents’ education level, the more favorable the oral self-care of their children [32]. Considering the relationship between socioeconomic factors and the prevalence of dental caries, the results suggest the necessity of a major educational intervention that focuses on parents with low levels of education. There are several oral health promotion programs that have been launched in other countries to control the disease and improve the oral health status of school children. Child oral health programs should be integrated with other public health sectors [33]. Among different preventive interventions, fluoride is found to be an effective and non-invasive agent for the management of dental caries, particularly water fluoridation and fluoride toothpaste, gel, and mouthwash, which are commonly incorporated into public health interventions to reduce the prevalence of dental caries [34]. Recently, data from the literature demonstrated that the oral health of children has improved because of the use of many preventive approaches, such as fluoride toothpaste, sealant treatment, and community water fluoridation programs [1].

Dentistry is facing an essential paradigm shift from a surgical specialty to a medical one; therefore, the non-operative treatment plans will become more and more central, and this project aimed to add to the pyramid of evidence. Therefore, taking into account the results of this survey, a new interventional program was drafted. The originality of the project lies in the vision of using fluoridated toothpaste not only as a means of preventing new carious lesions but also as a means of providing a therapeutic effect, which the study set out to evaluate. Starting from downstream oral health interventions, such as clinical prevention and oral health promotion, which can improve the quality of children’s lives, we aim to further develop the program towards upstream interventions. The program also involves a novel curriculum for dental hygienists aiming to promote efficient workforce models. The aim is to create primary care teams, including community health workers, in prevention and control and to build a supportive framework for integration of caries prevention and control into overall health initiatives. Caries can be considered a behavioral disease [21]. These behaviors differ according to the social environment, showing that underprivileged communities have worse oral health status. The present study proved that social context influenced the level of carious lesions but not the existence of restorations. Inequalities in socioeconomic status have been proven to play a determining role starting from preschool [35], and low social development could be correlated to a higher risk of caries prevalence [36]. It has also been stated that children coming from low-income families are more likely to suffer from increased levels of oral diseases during adulthood [37,38].

This relationship has not been a consistent finding in all studies. For example, in Brazil, the Human Development Index, average household income, and the number of public primary healthcare units were not associated with the proportion of preschool children with untreated decayed teeth in the poorer neighborhoods in southern Brazil. However, these indicators have been associated with the proportion of children with filled teeth living in richer areas [39,40].

Several psychosocial and materialistic concepts state that low socioeconomic families are exposed to higher health risks, including lack of access to care and worse environmental factors and living conditions [41]. From the psychosocial perspective, people living in underprivileged communities are more exposed to anxiety and stress, having an impact on the general quality of life, general health outcomes, and oral health.

Human relationships within the community are considered social capital, which is beneficial both to the individual as well as to the community [42]. An individual having high social capital also benefits from better health behaviors and receives more information from better resources, which can enhance their health status [41,42]. It has been shown that in communities with high social capital, people have a higher life expectancy, lower morbidity and mortality, and better health knowledge [41,42,43].

Regarding preventive hygiene behavior, dental floss was reported to be used more often by the Lithuanian children whose parents had a high educational level as compared with the children of the parents with a low educational level (26.1% vs. 14.8%, respectively) [44]. Moreover, oral health was better in those children who had regular dental check-ups and brushed their teeth from a young age [45].

The results from a Norwegian study since 2014 showed that though 12-year-old children were starting to behave independently, family background still influenced the children’s oral health behaviors and caries prevalence [46].

Limitations of the study include the fact that it is a cross-sectional study using a self-administered questionnaire, as this study design measures cause and effect at the same time. It is also possible to recognize some methodological difficulties in managing the data collection across the whole country. Furthermore, another limitation of this study is that the self-reported questionnaire does not include any question that evaluates the level of knowledge related to the purpose and the benefits of using fluoride.

The complex procedure of sample selection of both urban and rural schools and data processing, as well as the extensive calibration process, support the validity of the findings. The present study is the first in Romania to take caries prevalence as well as the involvement of major caries risk factors and oral health-related behaviors into consideration, utilizing the STEPS approach recommended by WHO guidelines. This allows for the evaluation of national trends as well as comparisons with other countries. The STEPS method recommends collecting data on a regular and ongoing basis.

## 5. Conclusions

In conclusion, our findings indicated that there was a correlation between the oral health status, preventive behavior, and educational level of the parents/caregivers of children in Romania. The results imply that increased caries prevalence may be caused by the low educational level of the parents. Additionally, socioeconomical county-level variables influenced caries prevalence and severity, being correlated to the presence of carious lesions and absence of treatments. The results described in detail the oral health status of children aged 11–14 years in Romania, providing baseline data for future research and demonstrating the socioeconomic impact on the prevalence of dental caries. Dental public health workers should not limit themselves to the collection of these data, but they should use later on the information from this survey for the design and implementation of prevention programs to improve the oral health situation of school children in a sustainable way.

## Figures and Tables

**Figure 1 ijerph-19-11102-f001:**
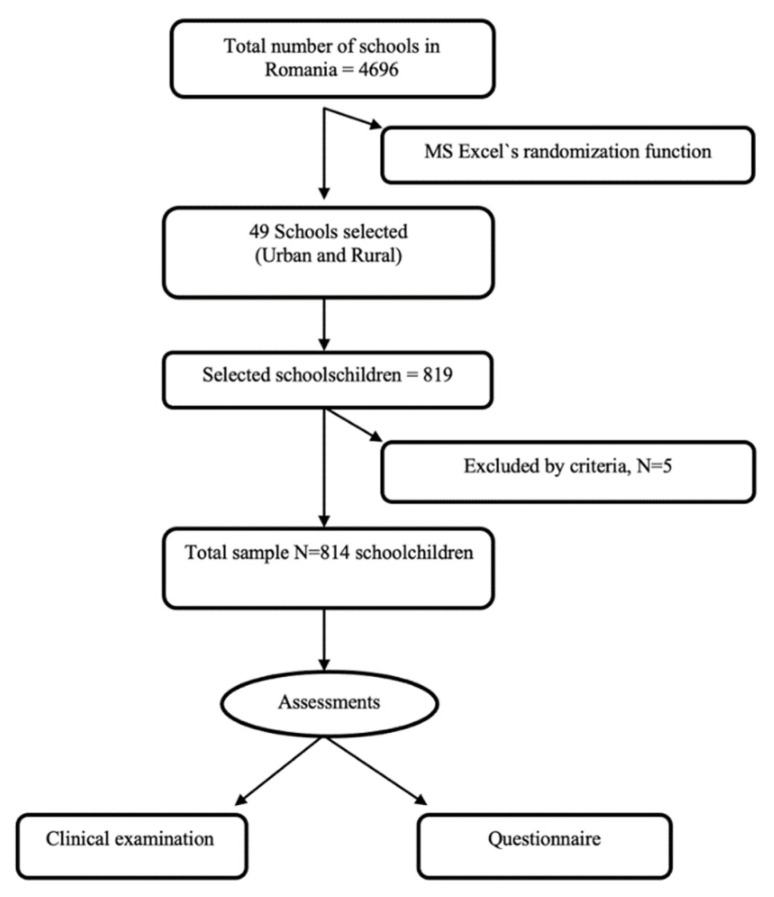
Flow diagram guideline of the sample selection.

**Table 1 ijerph-19-11102-t001:** Sample description according to gender and age.

Variable		N (%)
**Gender**	Male	388 (47.66)
	Female	426 (52.33)
**Age**	11 years	41 (5.01)
	12 years	514 (62.76)
	13 years	231 (28.21)
	14 years	24 (2.93)

**Table 2 ijerph-19-11102-t002:** Sample description according to mother’s and father’s education.

Type of Education	Mother’s Education (%)	Father’s Education (%)
**No school**	6 (0.73)	4 (0.49)
**Primary school (4th grade)**	20 (2.44)	24 (2.93)
**Gymnasium (8th grade)**	122 (14.90)	91 (11.11)
**Vocational school (10th grade)**	113 (13.80)	124 (15.14)
**High school (12th grade)**	234 (28.57)	268 (32.72)
**Post-secondary school**	44 (5.37)	29 (3.54)
**University studies**	181 (22.10)	149 (18.19)
**Don’t know/Don’t answer**	87 (10.62)	106 (12.94)
**I don’t live with the mother/father figure in the house**	3 (0.37)	16 (1.95)

**Table 3 ijerph-19-11102-t003:** Correlation between parent’s education and the preventive behavior.

Type of Preventive Behavior	Mother’s Education r_s_-rho (*p*, N)	Father’s Education r_s_-rho (*p*, N)
**Tooth pain/discomfort in last 12 months**	−0.01 ** (0.00, 814)	−0.03 (0.34, 814)
**Frequency of tooth brushing**	0.02 ** (0.00, 814)	0.18 ** (0.00, 814)
**Frequency of dental check-ups**	0.08 * (0.02, 730)	0.02 (0.57, 730)
**Oral hygiene aids—** **Toothbrush**	0.025 (0.04, 814)	0.03 (0.03, 814)
**Oral hygiene aids—Wo** **oden toothpicks**	−0.05 (0.14, 814)	−0.06 (0.05, 814)
**Oral hygiene aids—** **plastic toothpicks**	−0.03 (0.37, 814)	−0.03 (0.32, 814)
**Oral hygiene aids—** **dental floss**	0.02 (0.40, 814)	0.02 (0.43, 814)
**Oral hygiene aids—something else**	0.06 ** (0.00, 27)	0.03 (0.08, 27)

* *p* < 0.05, ** *p* < 0.01.

**Table 4 ijerph-19-11102-t004:** Correlation between parents’ education and the clinical status.

Type of Index	Mother ‘s Education r_s_-rho(*p*, N)	Father’s Education r_s_-rho (*p*, N)
MT	−0.01 (0.64, 814)	−0.00 (0.89, 814)
D_3_T	−0.19 ** (0.00, 814)	−0.18 ** (0.00, 814)
RT	0.09 ** (0.00, 814)	0.06 (0.07, 814)
DMFT	−0.09 ** (0.00, 814)	−0.10 ** (0.00, 814)

** *p* < 0.01.

**Table 5 ijerph-19-11102-t005:** Correlation between different types of preventive behaviors and MT, D_3_T, RT, and DMFT indices.

Type of Behaviors	MT r_s_-rho (*p*, N)	D_3_T r_s_-rho (*p*, N)	RT r_s_-rho (*p*, N)	DMFT r_s_-rho (*p*, N)
**Tooth pain/tooth discomfort in last 12 months**	0.09 ** (0.00, 814)	0.15 ** (0.00, 814)	0.03 (0.36, 814)	0.11 ** (0.00, 814)
**Frequency of tooth brushing**	0.00 (0.85, 814)	−0.01 ** (0.00, 814)	0.07 * (0.03, 814)	−0.063 (0.071, 814)
**Frequency of dental check-ups**	−0.07 * (0.03, 814)	−0.02 (0.54, 730)	0.00 (0.82, 730)	0.02 (0.55, 730)
**Oral hygiene aids—** **Toothbrush**	0.00 (0.92, 814)	0.03 (0.92, 814)	0.00 (0.99, 814)	−0.03 (0.33, 814)
**Oral hygiene aids—Wo** **oden toothpicks**	0.02 (0.52, 814)	0.00 (0.97, 814)	−0.04 (0.19, 814)	0.02 (0.48, 814)
**Oral hygiene aids—** **plastic toothpicks**	0.13 ** (0.00, 814)	−0.33 (0.35, 814)	−0.02 (0.42, 814)	−0.00 (0.80, 814)
**Oral hygiene aids—** **dental floss**	0.02 (0.46, 814)	−0.08 * (0.02, 814)	0.03 (0.38, 814)	−0.01 (0.66, 814)
**Oral hygiene aids—something else**	−0.04 (0.81, 27)	−0.29 (0.12, 27)	−0.01 (0.95, 27)	−0.22 (0.26, 27)

* *p* < 0.05, ** *p* < 0.01.

**Table 6 ijerph-19-11102-t006:** Correlation between the MT, D_3_T, and RT indices and the development index of the county.

Type of Index	County Development Index (LHDI 2011)
MT	−0.012 (0.0723, N = 814)
D_3_T	−0.070 * (0.044, N = 814)
RT	−0.064 (0.068, N = 814)

* *p* < 0.05.

## Data Availability

The data presented in this study are available on request from the corresponding author. The data are not publicly available in accordance with consent provided by participants on the use of confidential data.

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
