# Peer review of "The Impact of Parental Education on Schoolchildren’s Oral Health—A Multicenter Cross-Sectional Study in Romania"

_ijerph, 2022, doi:10.3390/ijerph191711102_

Round 1
Reviewer 1 Report
The authors did a thourough study, but there is much unclear about the participation rate. This has to be added with much more detail. Now, it seems that the classes are small in Romenia (<20 per class) and the children who participated are pretty old (I would expect most will be between 11 and 12). Furthermore, the number of parents who visited gymnasium etc was very high as this is mostly the highest grade after primary school. Probably, this is another system in Romenia that in the Western world, but this has to be explained so that the data can be analyzed taking this into perspective. Also, I do not see strong correlations but at most very weak, although significant correlations. Finally, the authors tested many possible correlations, but was corrected for multiple testing.
Reviewer 2 Report
Conducting this study was a huge undertaking and the authors should be acknowledged for that. However, there are some serious limitations. The authors note that dental caries is almost entirely preventable [line53], yet there is no explanation about what prevents this disease. Evidence suggests STRONGLY that the appropriate use of fluorides and pit and fissure sealants prevent this disease. Although the first reference is about the use of fluoride to prevent caries, NOT one word about fluoride is contained in this manuscript. For example, what is the F level in most drinking water in Romania AND do the toothpastes contain fluoride? Are dental sealants available in Romania? Frequent visits to the dentist [line 70-71] is not usually a preventive measure, except when operator F is being applied. Level of parental education is, indeed, important regarding the oral health of their children BUT other factors also are pivotal.
The authors of this manuscript must tell the reader what primary preventive agents/regimens are available in Romania. Is community water fluoridation available and or what is the F level of drinking water?
Also, authors should consider explaining that one of the serious limitations of the study is that parents were not asked questions similar to:
Have you ever heard of fluoride?
What is the purpose of fluoride?
What do you or your child put on the toothbrush when they brush? If it is toothpaste, does it contain fluoride?
Another limitation of the study is that the 10 trained examiners di not conduct re-examination of a certain number of children during the study.
Individuals with low levels of education likely have low paying jobs and UNLIKELY to be able to take their child to a dentist or buy toothbrushes or toothpaste.
Round 2
Reviewer 1 Report
The paper has improved.
Reviewer 2 Report
Dear Authors, I gather you have tried your best to make changes based on my critique. The single most glaring deficiency now include the discussion and conclusions. I thought you would incorporate into both approaches to making change. While the report does 'describe in detail the oral health status of children aged 11-14 years in Romania, providing baseline data for future research, demonstrating socioeconomic impact on the prevalence of dental caries.' it does nothing to suggest what might be done to prevent dental caries. Your data suggest strongly that a major educational intervention be implemented that focuses on parents with low levels of education. There are all kinds of school based programs [fluoride mouth rinse, dental sealants, fluoride tablets] that could be considered. You should use these data to design and implement these kinds of programs. Dental public health workers should not be doing these kinds of surveys simply to provide baseline data for future research, as the conclusions indicate.
